# Set-based Meta-Interpolation for Few-Task Meta-Learning

**Seanie Lee**[1]*, **Bruno Andreis**[1]*,
**Kenji Kawaguchi** [2], **Juho Lee**[1,3], **Sung Ju Hwang**[1]
KAIST[1], National University of Singapore[2], AITRICS[3]
{lsnfamily02, andries}@kaist.ac.kr,
kenji@comp.nus.edu.sg, {juholee, sjhwang82}@kaist.ac.kr

## Abstract

Meta-learning approaches enable machine learning systems to adapt to new tasks given few examples by leveraging knowledge from related tasks. However, a large number of meta-training tasks are still required for generalization to unseen tasks during meta-testing, which introduces a critical bottleneck for real-world problems that come with only few tasks, due to various reasons including the difficulty and cost of constructing tasks. Recently, several task augmentation methods have been proposed to tackle this issue using domain-specific knowledge to design augmentation techniques to densify the meta-training task distribution. However, such reliance on domain-specific knowledge renders these methods inapplicable to other domains. While Manifold Mixup based task augmentation methods are domain-agnostic, we empirically find them ineffective on non-image domains. To tackle these limitations, we propose a novel domain-agnostic task augmentation method, Meta-Interpolation, which utilizes expressive neural set functions to densify the meta-training task distribution using bilevel optimization. We empirically validate the efficacy of Meta-Interpolation on eight datasets spanning across various domains such as image classification, molecule property prediction, text classification and sound classificattion. Experimentally, we show that Meta-Interpolation consistently outperforms all the relevant baselines. Theoretically, we prove that task interpolation with the set function regularizes the meta-learner to improve generalization.

## 1  Introduction

The ability to learn a new task given only a few examples is crucial for artificial intelligence. Recently, meta-learning [39, 3] has emerged as a viable method to achieve this objective and enables machine learning systems to quickly adapt to a new task by leveraging knowledge from other related tasks seen during meta-training. Although existing meta-learning methods can efficiently adapt to new tasks with few data samples, a large dataset of meta-training tasks is still required to learn meta-knowledge that can be transferred to unseen tasks. For many real-world applications, such extensive collections of meta-training tasks may be unavailable. Such scenarios give rise to the few-task meta-learning problem where a meta-learner can easily memorize the meta-training tasks but fail to generalize well to unseen tasks. The few-task meta-learning problem usually results from the difficulty in task generation and data collection. For instance, in the medical domain, it is infeasible to collect a large amount of data to construct extensive meta-training tasks due to privacy concerns. Moreover, for natural language processing, it is not straightforward to split a dataset into tasks, and hence entire datasets are treated as tasks [30].

---

*Equal Contribution. Order of the authors was determined by a coin toss.

36th Conference on Neural Information Processing Systems (NeurIPS 2022).

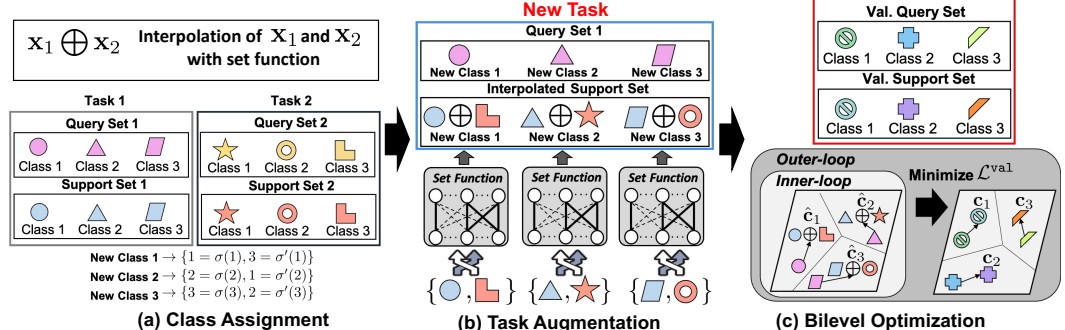

Figure 1: **Concept.** Three-way one-shot classification problem. **(a)** A new class is assigned to a pair of classes sampled without replacement from the pool of meta-training tasks. **(b)** The support sets are interpolated with a set function and paired with a query set. **(c)** Bilevel optimization of the set function and meta-learner.

Several works have been proposed to tackle the few-task meta-learning problem using task augmentation techniques such as clustering a dataset into multiple tasks [30], leveraging strong image augmentation methods such as vertical flipping to construct new classes [32], and the employment of Manifold Mixup [44] for densifying the meta-training task distribution [49, 50]. However, majority of these techniques require domain-specific knowledge to design such task augmentations and hence cannot be applied to other domains. While Manifold Mixup based methods [49, 50] are domain-agnostic, we empirically find them ineffective for mitigating meta-overfitting in few-task meta-learning especially in non-image domains such as chemical and text, and that they sometimes degrade generalization performance.

In this work, we focus solely on *domain-agnostic* task augmentation methods that can densify the meta-training task distribution to prevent meta-overfitting and improve generalization at meta-testing for few-task meta-learning. To tackle the limitations already discussed, we propose a novel domain-agnostic task augmentation method for metric based meta-learning models. Our method, Meta-Interpolation, utilizes expressive neural set functions to interpolate two tasks and the set functions are trained with bilevel optimization so that a meta-learner trained on the interpolated tasks generalizes to tasks in the meta-validation set. As a consequence of end-to-end training, the learned augmentation strategy is tailored to each specific domain without the need for specialized domain knowledge.

For example, for $K$-way classification, we sample two tasks consisting of support and query sets and assign a new class $k$ to each pair of classes $\{\sigma(k), \sigma'(k)\}$ for $k = 1, \ldots, K$, where $\sigma, \sigma'$ are permutations on $\{1, \ldots, K\}$ as depicted in Figure 1a. Hidden representations of the support set with classes $\sigma(k)$ and $\sigma'(k)$ are then transformed into a single support set using a set function that maps a set of two vectors to a single vector. We refer to the output of the set function as the interpolated support set and these are used to compute class prototypes. As shown in Figure 1b, the interpolated support set is paired with a query set (query set 1 in Figure 1a)), randomly selected from the two tasks to obtain a new task. Lastly, we optimize the set function so that a meta-learner trained on the augmented task can minimize the loss on the meta-validation tasks as illustrated in Figure 1c.

To verify the efficacy of our method, we empirically show that it significantly improves the performance of Prototypical Networks [40] on the few-task meta-learning problem across multiple domains. Our method outperforms the relevant baselines on eight few-task meta-learning benchmark datasets spanning image classification, chemical property prediction, text classification, and sound classification. Furthermore, our theoretical analysis shows that our task interpolation method with the set function regularizes the meta-learner and improves generalization performance.

Our contribution is threefold:

- We propose a novel domain-agnostic task augmentation method, Meta-Interpolation, which leverages expressive set functions to densify the meta-training task distribution for the few-task meta-learning problem.

- We theoretically analyze our model and show that it regularizes the meta-learner for better generalization.

- Through extensive experiments, we show that Meta-Interpolation significantly improves the performance of Prototypical Network on various domains such as image, text, and chemical molecule, and sound classification on the few-task meta-learning problem.

## 2 Related Work

**Meta-Learning** The two mainstream approaches to meta-learning are gradient based [10, 33, 14, 24, 12, 36, 37] and metric based meta-learning [45, 40, 42, 29, 26, 6, 38]. The former formulates meta-knowledge as meta-parameters such as the initial model parameters and performs bilevel optimization to estimate the meta-parameters so that a meta-learner can generalize to unseen tasks with few gradient steps. The latter learns an embedding space where classification is performed by measuring the distance between a query and a set of class prototypes. In this work, we focus on metric based meta-learning with *fewer number* of meta-training tasks, i.e., few-task meta-learning. We propose a novel task augmentation method that densifies the meta-training task distribution and mitigates overfitting due to the fewer number of meta-training tasks for better generalization to unseen tasks.

**Task Augmentation for Few-Task Meta-learning** Several methods have been proposed to augment the number of meta-training tasks to mitigate overfitting in the context of few-task meta-learning. Ni et al. [32] apply strong data augmentations such as vertical flip to images to create a new class. For text classification, Murty et al. [30] split meta-training tasks into latent reasoning categories by clustering data with a pretrained language model. However, they require domain-specific knowledge to design such augmentations, and hence the resulting augmentation techniques are inapplicable to other domains where there is no well-defined data augmentation or pretrained model. In order to tackle this limitation, Manifold Mixup-based task augmentations have also been proposed. MetaMix [49] interpolates support and query sets with Manifold Mixup [44] to construct a new query set. MLTI [50] performs Manifold Mixup [44] on support and query sets from two tasks for task augmentation. Although these methods are domain-agnostic, we empirically find that they are not effective in some domains and can degrade generalization performance. In contrast, we propose to train an expressive neural set function to interpolate two tasks with bilevel optimization to find optimal augmentation strategies tailored specifically to each domain.

## 3 Method

**Preliminaries** In meta-learning, we are given a finite set of tasks $\{\mathcal{T}_t\}_{t=1}^T$, which are i.i.d samples from an unknown task distribution $p(\mathcal{T})$. Each task $\mathcal{T}_t$ consists of a support set $\mathcal{D}_t^s = \{(\mathbf{x}_{t,i}^s, y_{t,i}^s)\}_{i=1}^{N_s}$ and a query set $\mathcal{D}_t^q = \{(\mathbf{x}_{t,i}^q, y_{t,i}^q)\}_{i=1}^{N_q}$, where $\mathbf{x}_{t,i}$ and $y_{t,i}$ denote a data point and its corresponding label respectively. Given a predictive model $\hat{f}_{\theta,\lambda} \coloneqq f_{\theta_L}^L \circ \cdots \circ f_{\theta_{l+1}}^{l+1} \circ \varphi_\lambda \circ f_{\theta_l}^l \circ \cdots \circ f_{\theta_1}^1$ with $L$ layers, we want to estimate the parameter $\theta$ that minimizes the meta-training loss and generalizes to query sets $\mathcal{D}_*^q$ sampled from an unseen task $\mathcal{T}_*$ using the support set $\mathcal{D}_*^s$, where $\lambda$ is a hyperparameter for the function $\varphi_\lambda$. In this work, we primarily focus on metric based meta-learning methods rather than gradient based meta-learning methods due to efficiency and empirically higher performance over the gradient based methods on the tasks we consider.

**Problem Statement** In this work, we focus solely on *few-task* meta-learning. Here, the number of meta-training tasks drawn from the meta-training distribution is extremely small and the goal of a meta-learner is to learn meta-knowledge from such limited tasks that can be transferred to unseen tasks during meta-testing. The key challenges here are preventing the meta-learner from overfitting on the meta-training tasks and generalizing to unseen tasks drawn from a meta-test set.

**Metric Based Meta-Learning** The goal of metric based meta-learning is to learn an embedding space induced by $\hat{f}_{\theta,\lambda}$, where we perform classification by computing distances between data points and class prototypes. We adopt Prototypical Network (ProtoNet) [40] for $\hat{f}_{\theta,\lambda}$, where $\varphi_\lambda$ is the identity function. Specifically, for each task $\mathcal{T}_t$ with its corresponding support $\mathcal{D}_t^s$ and query $\mathcal{D}_t^q$ sets, we compute class prototypes $\{\mathbf{c}_k\}_{k=1}^K$ as the average of the hidden representation of the support samples belonging to the class $k$ as follows:

$$\mathbf{c}_k \coloneqq \frac{1}{N_k} \sum_{\substack{(\mathbf{x}_{t,i}^s, y_{t,i}^s) \in \mathcal{D}_t^s \\ y_{t,i}=k}} \hat{f}_{\theta,\lambda}(\mathbf{x}_{t,i}^s) \in \mathbb{R}^D \tag{1}$$

where $N_k$ denotes the number of instances belonging to the class $k$. Given a metric $d(\cdot, \cdot) : \mathbb{R}^D \times \mathbb{R}^D \mapsto \mathbb{R}$, we compute the probability of a query point $\mathbf{x}_{t,i}^q$ being assigned to the class $k$ by measuring the distance between the hidden representation $\hat{f}_{\theta,\lambda}(\mathbf{x}_{t,i}^q)$ and the class prototype $\mathbf{c}_k$ followed by

| **Algorithm 1** Meta-training | **Algorithm 2** HyperGrad [27] |
|---|---|
| **Require:** Tasks $\{\mathcal{T}_t^{\text{train}}\}_{t=1}^T$ $\{\mathcal{T}_t^{\text{val}'}\}_{t'=1}^{T'}$, learning rate $\alpha, \eta \in \mathbb{R}_+$, update period $S$, and batch size $B$. | **Require:** model parameter $\theta$, hyperparamter $\lambda$, validation tasks $\{T_{t'}^{\text{val}}\}_{t'=1}^{T'}$, learning rate $\alpha$, gradient of training loss w.r.t $\theta$ $\frac{\partial \mathcal{L}_{tr}}{\partial \theta}$, batch size $B'$, and the number of iterations for Neumann series $q \in \mathbb{N}$. |

Algorithm 1 Meta-training:
1: Initialize parameters $\theta, \lambda$
2: **for all** $i \leftarrow 1, \ldots, M$ **do**
3: $\quad \mathcal{L}_{tr} \leftarrow 0$
4: $\quad$ **for all** $j \leftarrow 1, \ldots, B$ **do**
5: $\quad\quad$ Sample two tasks $\mathcal{T}_{t_1} = \{\mathcal{D}_{t_1}^s, \mathcal{D}_{t_1}^q\}$ and $\mathcal{T}_{t_2} = \{\mathcal{D}_{t_2}^s, \mathcal{D}_{t_2}^q\}$ from $\{\mathcal{T}_t^{\text{train}}\}_{t=1}^T$.
6: $\quad\quad \hat{\mathcal{D}}^s \leftarrow \text{Interpolate}(\mathcal{D}_{t_1}^s, \mathcal{D}_{t_2}^s, \varphi_\lambda)$ with Eq.3.
7: $\quad\quad \hat{\mathcal{T}} \leftarrow \{\hat{\mathcal{D}}^s, \mathcal{D}_{t_1}^q\}$
8: $\quad\quad \mathcal{L}_{tr} += \frac{1}{2B} \mathcal{L}_{\text{singleton}}(\lambda, \theta, \mathcal{T}_{t_1})$
9: $\quad\quad \mathcal{L}_{tr} += \frac{1}{2B} \mathcal{L}_{\text{mix}}(\lambda, \theta, \hat{\mathcal{T}})$
10: $\quad$ **end for**
11: $\quad \theta \leftarrow \theta - \alpha \frac{\partial \mathcal{L}_{tr}}{\partial \theta}$
12: $\quad$ **if** $\text{mod}(i, S) = 0$ **then**
13: $\quad\quad g \leftarrow \text{HyperGrad}(\theta, \lambda, \{\mathcal{T}_{t'}^{\text{val}}\}_{t'=1}^{T'}, \alpha, \frac{\partial \mathcal{L}_{tr}}{\partial \theta})$
14: $\quad\quad \lambda \leftarrow \lambda - \eta \cdot g$
15: $\quad$ **end if**
16: **end for**
17: **return** $\theta, \lambda$

Algorithm 2 HyperGrad [27]:
1: $\mathcal{L}_V \leftarrow 0$
2: **for all** $i \leftarrow 1, \ldots, B'$ **do**
3: $\quad$ Sample a task $\mathcal{T}_t$ from $\{\mathcal{T}_{t'}^{\text{val}}\}_{t'=1}^{T'}$.
4: $\quad \mathcal{L}_V += \frac{1}{B'} \mathcal{L}_{\text{singleton}}(\lambda, \theta; \mathcal{T})$
5: **end for**
6: $\mathbf{v}_1 \leftarrow \frac{\partial \mathcal{L}_V}{\partial \theta}$
7: Initialize $\mathbf{p} \leftarrow \text{deepcopy}(\mathbf{v}_1)$
8: **for all** $j \leftarrow 1, \ldots, q$ **do**
9: $\quad \mathbf{v}_1 -= \alpha \cdot \text{grad}(\frac{\partial \mathcal{L}_{tr}}{\partial \theta}, \theta, \text{grad\_outputs} = \mathbf{v}_1)$
10: $\quad \mathbf{p} += \mathbf{v}_1$
11: **end for**
12: $\mathbf{v}_2 \leftarrow \text{grad}(\frac{\mathcal{L}_{tr}}{\partial \theta}, \lambda, \text{grad\_outputs} = \alpha\mathbf{p})$
13: **return** $\frac{\partial \mathcal{L}_V}{\partial \lambda} - \mathbf{v}_2$
14:
15:

softmax. With the class probability, we compute the cross-entropy loss for ProtoNet as follows:

$$\mathcal{L}_{\text{singleton}}(\lambda, \theta; \mathcal{T}_t) := \sum_{i,k} \mathbb{1}_{\{y_{t,i}=k\}} \cdot \log \frac{\exp(-d(\hat{f}_{\theta,\lambda}(\mathbf{x}_{t,i}^q), \mathbf{c}_k))}{\sum_{k'} \exp(-d(\hat{f}_{\theta,\lambda}(\mathbf{x}_{t,i}^q), \mathbf{c}_{k'}))} \tag{2}$$

where $\mathbb{1}$ is an indicator function. At meta-test time, a test query is assigned a label based on the minimal distance to a class prototype, i.e., $y_*^q = \arg\min_k d(\hat{f}_{\theta,\lambda}(\mathbf{x}_*^q), \mathbf{c}_k)$. However, optimizing $\frac{1}{T} \sum_{t=1}^T \mathcal{L}_{\text{singleton}}(\lambda, \theta; \mathcal{T}_t)$ w.r.t $\theta$ is prone to overfitting since we are given only a small number of meta-training tasks. The meta-learner tends to memorize the meta-training tasks, which limits its generalization to new tasks at meta-test time [51, 35].

**Meta-Interpolation for Task Augmentation** In order to tackle the meta-overfitting problem with a small number of tasks, we propose a novel data-driven domain-agnostic task augmentation framework which enables the meta-learner trained on few tasks to generalize to unseen few-shot classification tasks. Several methods have been proposed to densify the meta-training tasks. However, they heavily depend on the augmentation of images [32] or need a pretrained language model for task augmentation [30]. Although Manifold Mixup based methods [49, 50] are domain-agnostic, we empirically find them ineffective in certain domains. Instead, we optimize expressive neural set functions to augment tasks to enhance the generalization of a meta-learner to unseen tasks. As a consequence of end-to-end training, the learned augmentation strategy is tailored to each domain.

Specifically, let $\varphi_\lambda : \mathbb{R}^{n \times d} \to \mathbb{R}^d$ be a set function which maps a set of $d$ dimensional vectors with cardinality $n$ to a $d$ dimensional vector. In all our experiments, we use Set Transformer [23] for $\varphi_\lambda$. Given a pair of tasks $\mathcal{T}_{t_1} = \{\mathcal{D}_{t_1}^s, \mathcal{D}_{t_1}^q\}$ and $\mathcal{T}_{t_2} = \{\mathcal{D}_{t_2}^s, \mathcal{D}_{t_2}^q\}$ with corresponding support and query sets for $K$ way classification, we construct new classes by choosing $K$ pairs of classes from the two tasks. We sample permutations $\sigma_{t_1}$ and $\sigma_{t_2}$ on $\{1, \ldots, K\}$ for each task $\mathcal{T}_{t_1}$ and $\mathcal{T}_{t_2}$ respectively and assign class $k$ to the pair $\{\sigma_{t_1}(k), \sigma_{t_2}(k)\}$ for $k = 1, \ldots, K$. For the newly assigned class $k$, we pair two instances from classes $\sigma_{t_1}(k)$ and $\sigma_{t_2}(k)$ and interpolate their hidden representations with the set function $\varphi_\lambda$. The class prototypes for class $k$ are computed using the output of $\varphi_\lambda$ as follows:

$$S_k := \{(\{\mathbf{x}_{t_1,i}^s, \mathbf{x}_{t_2,j}^s\}, k) \mid (\mathbf{x}_{t_1,i}^s, y_{t_1,i}^s) \in \mathcal{D}_{t_1}^s, y_{t_1,i}^s = \sigma_{t_1}(k), (\mathbf{x}_{t_2,j}^s, y_{t_2,j}^s) \in \mathcal{D}_{t_2}^s, y_{t_2,j}^s = \sigma_{t_2}(k)\}$$

$$\mathbf{h}_{t_1,i}^{s,l} := (f_{\theta_l}^l \circ \cdots f_{\theta_1}^1)(\mathbf{x}_{t_1,i}^s), \quad \mathbf{h}_{t_2,j}^{s,l} := (f_{\theta_l}^l \circ \cdots f_{\theta_1}^1)(\mathbf{x}_{t_2,j}^s) \in \mathbb{R}^d$$

$$\hat{\mathbf{c}}_k := \frac{1}{|S_k|} \sum_{(\{\mathbf{x}_{t_1,i}^s, \mathbf{x}_{t_2,j}^s\}, k) \in S_k} \left( f_{\theta_L}^L \circ \cdots \circ f_{\theta_{l+1}}^{l+1} \right) \left( \varphi_\lambda(\{\mathbf{h}_{t_1,i}^{s,l}, \mathbf{h}_{t_2,j}^{s,l}\}) \right) \in \mathbb{R}^D$$

$$\hat{\mathcal{D}}^s := \{\hat{\mathbf{c}}_1, \ldots, \hat{\mathbf{c}}_K\}$$

(3)

where we define $\hat{\mathcal{D}}^s$ to be the set of all the interpolated prototypes $\hat{\mathbf{c}}_k$ for $k = 1, \ldots, K$. For queries, we do not perform any interpolation. Instead, we use $\mathcal{D}_{t_1}^q$ as the query set and compute its hidden

representation $\hat{f}_{\theta,\lambda}(\mathbf{x}_{t_1,i}^q) \in \mathbb{R}^D$. We then measure the distance between the query with $y_{t_1,i}^q = \sigma_{t_1}(k)$ and the interpolated prototype of class $k$ to compute the loss as follows:

$$\mathcal{L}_{\mathrm{mix}}(\lambda, \theta, \hat{\mathcal{T}}) := -\sum_{i,k} \mathbb{1}_{\{y_{t_1,i}^q = \sigma_{t_1}(k)\}} \cdot \log \frac{\exp(-d(\hat{f}_{\theta,\lambda}(\mathbf{x}_{t_1,i}^q), \hat{\mathbf{c}}_k))}{\sum_{k'} \exp(-d(\hat{f}_{\theta,\lambda}(\mathbf{x}_{t_1,i}^q), \hat{\mathbf{c}}_{k'}))} \tag{4}$$

where $\hat{\mathcal{T}} = \{\hat{\mathcal{D}}^s, \mathcal{D}_{t_1}^q\}$. The intuition behind interpolating only support sets is to construct harder tasks that a meta-learner cannot memorize. Alternatively, we can interpolate only query sets. However, this is computationally more expensive since the size of query sets is usually larger than that of support sets. In Section 5, we empirically show that interpolating either support or query sets achieves higher training loss than interpolating both, which empirically supports the intuition. Lastly, we also use the original task $\mathcal{T}_{t_1}$ to evaluate the loss $\mathcal{L}_{\mathrm{singleton}}(\lambda, \theta, \mathcal{T}_{t_1})$ in Eq. 2 by passing the corresponding support and query set to $\hat{f}_{\theta,\lambda}$. The additional forward pass enriches the diversity of the augmented tasks and makes meta-training consistent with meta-testing since we do not perform any task augmentation in the meta-testing stage.

**Optimization** Since jointly optimizing $\theta$, the parameters of ProtoNet, and $\lambda$, the parameters of the set function $\varphi_\lambda$, with few tasks is prone to overfitting, we consider $\lambda$ as hyperparameters and perform bilevel optimization with meta-training and meta-validation tasks as follows:

$$\lambda^* := \arg\min_{\lambda} \frac{1}{T'} \sum_{t=1}^{T'} \mathcal{L}_{\mathrm{singleton}}(\lambda, \theta^*(\lambda); \mathcal{T}_t^{\mathrm{val}}) \tag{5}$$

$$\theta^*(\lambda) := \arg\min_{\theta} \frac{1}{2T} \sum_{t=1}^{T} \mathcal{L}_{\mathrm{singleton}}(\lambda, \theta; \mathcal{T}_t^{\mathrm{train}}) + \mathcal{L}_{\mathrm{mix}}(\lambda, \theta; \hat{\mathcal{T}}_t) \tag{6}$$

where $\mathcal{T}_t^{\mathrm{train}}, \mathcal{T}_t^{\mathrm{val}}, \hat{\mathcal{T}}_t$ denote the meta-training, meta-validation, and interpolated task, respectively. Since computing the exact gradient w.r.t $\lambda$ is intractable due to the long inner optimization steps in Eq. 6, we leverage the implicit function theorem to approximate the gradient as Lorraine et al. [27]. Moreover, we alternately update $\theta$ and $\lambda$ for computational efficiency as described in Algo. 1 and 2.

## 4   Theoretical Analysis

In this section, we theoretically investigate the behavior of the Set Transformer and how it induces a distribution dependent regularization, which is then shown to have the ability to control the Rademacher complexity for better generalization. To analyze the behavior of the Set Transformer, we first define it concretely with the attention mechanism $A(Q, K, V) = \mathrm{softmax}(\sqrt{d^{-1}}QK^\top)V$. Given $h, h' \in \mathbb{R}^d$, define $H_1^{\{h,h'\}} = [h, h']^\top \in \mathbb{R}^{2 \times d}$ and $H_1^{\{h\}} = h^\top \in \mathbb{R}^{1 \times d}$. Then, for any $r \in \{\{h, h'\}, \{h\}\}$, the output of the Set Transformer $\varphi_\lambda(r)$ is defined as follows:

$$\varphi_\lambda(r) = A(Q_2, K_2^r, V_2^r)^\top \in \mathbb{R}^d, \tag{7}$$

where $Q_2 = SW_2^Q + b_2^Q$, $Q_1^r = H_1^r W_1^Q + \mathbf{1}_2 b_1^Q$, $K_j^r = H_j^r W_j^K + \mathbf{1}_2 b_j^K$, $V_j^r = H_j^r W_j^V + \mathbf{1}_2 b_j^V$ (for $j \in \{1, 2\}$), and $H_2^r = A(Q_1^r, K_1^r, V_1^r) \in \mathbb{R}^{n \times d}$. $Q_j, K_j, V_j$ denote query, key, and value for the attention mechanism for $j = 1, 2$, respectively. Here, $\mathbf{1}_2 = [1, \ldots, 1]^\top \in \mathbb{R}^n$, $W_j^Q, W_j^K, W_j^V \in \mathbb{R}^{d \times d}$, $b_j^Q, b_j^K, b_j^V \in \mathbb{R}^{1 \times d}$, $Q_1^r, K_j^r, V_j^r \in \mathbb{R}^{n \times d}$, and $Q_2 \in \mathbb{R}^{1 \times d}$. Let $l \in \{1, \ldots, L\}$.

Our analysis will show the importance of the following quantity of the Set Transformer in our method:

$$\alpha_{ij}^{(t,t')} = p_2^{(t,t',i,j)}(1 - p_1^{(t,t',i,j)}) + (1 - p_2^{(t,t',i,j)})(1 - \tilde{p}_1^{(t,t',i,j)}), \tag{8}$$

where $p_1^{(t,t',i,j)} = \mathrm{softmax}(\sqrt{d^{-1}}Q_1^{\{h_{t,i}, h_{t',j}\}}(K_1^{\{h_{t,i}, h_{t',j}\}})^\top)_{1,1}$, $\tilde{p}_1^{(t,t',i,j)} = \mathrm{softmax}(\sqrt{d^{-1}}Q_1^{\{h_{t,i}, h_{t',j}\}}(K_1^{\{h_{t,i}, h_{t',j}\}})^\top)_{2,1}$, $p_2^{(t,t',i,j)} = \mathrm{softmax}(\sqrt{d^{-1}}Q_2(K_2^{\{h_{t,i}, h_{t',j}\}})^\top)_{1,1}$ with $h_{t,i} = \phi_\theta^l(\mathbf{x}_{t,i}^s)$ and $\phi_\theta^l = f_{\theta_l}^l \circ \cdots \circ f_{\theta_1}^1$. For a matrix $A \in \mathbb{R}^{m \times n}$, $A_{i,j}$ denotes the entry for $i$-th row and $j$-th column of the matrix $A$.

We now introduce the additional notation and problem setting to present our results. Define $W = (W_1^V W_2^V)^\top \in \mathbb{R}^{d \times d}$, $b = (b_1^V W_2^V + b_2^V)^\top \in \mathbb{R}^d$, $L_t(\mathbf{c}) = -\frac{1}{n} \sum_{i=1}^{n} \log \frac{\exp(-d(\hat{f}_{\theta,\lambda}(\mathbf{x}_{t,i}^q), \mathbf{c}_{y_{t,i}^q}))}{\sum_{k'} \exp(-d(\hat{f}_{\theta,\lambda}(\mathbf{x}_{t,i}^q), \mathbf{c}_{k'}))}$,

and $I_{t,k} = \{i \in [N_s^{(t)}] : y_{t,i}^s = k\}$, where $N_s^{(t)} = |\mathcal{D}_t^s|$. We also define the empirical measure $\mu_{t,k} = \frac{1}{|I_{t,k}|} \sum_{i \in I_{t,k}} \delta_i$ over the index $i \in [N_s^{(t)}]$ with the Dirac measures $\delta_i$. Let $U[K]$ be the uniform distribution over $\{1, \ldots, K\}$. For any function $\varphi$ and point $u$ in its domain, we define the $j$-th order tensor $\partial^j \varphi(u) \in \mathbb{R}^{d \times d \times \cdots \times d}$ by $\partial^j \varphi(u)_{i_1 i_2 \cdots i_j} = \frac{\partial^j}{\partial u_{i_1} u_{i_2} \cdots \partial u_{i_j}} \varphi(u)$. For example, $\partial^1 \varphi(u)$ and $\partial^2 \varphi(u)$ are the gradient and the Hessian of $\varphi$ evaluated at $u$. For any $j$-th order tensor $\partial^j \varphi(u)$, we define the vectorization of the tensor by $\text{vec}[\partial^j \varphi(u)] \in \mathbb{R}^{d^j}$. For an vector $a \in \mathbb{R}^d$, we define $a^{\otimes j} = a \otimes a \otimes \cdots \otimes a \in \mathbb{R}^{d^j}$ where $\otimes$ represents the Kronecker product. We assume that $\partial^r g \left( W \phi_{\theta_l}^l(\mathbf{x}_{t,i}^s) + b \right) = 0$ for all $r \geq 2$, where $g := f_{\theta_L}^L \circ \cdots \circ f_{\theta_{l+1}}^{l+1}$. This assumption is satisfied, for example, if $g$ represents a deep neural network with ReLU activations. This assumption is also satisfied in the simpler special case considered in the proposition below.

The following theorem shows that $\mathcal{L}_{\text{mix}}(\lambda, \theta, \hat{\mathcal{T}}_{t,t'})$ is approximately $\mathcal{L}_{\text{singleton}}(\lambda, \theta; \mathcal{T}_t)$ plus regularization terms on the directional derivatives of $\phi_{\theta_l}^l$ on the direction of $W(\phi_{\theta_l}^l(\mathbf{x}_{t',j}^s) - \phi_{\theta_l}^l(\mathbf{x}_{t,i}^s))$:

**Theorem 1.** *For any $J \in \mathbb{N}_+$, if $c \mapsto d(y, c)$ is $J$-times differentiable for all $y$, then the $J$-th order approximation of $\mathcal{L}_{mix}(\lambda, \theta, \hat{\mathcal{T}}_{t,t'})$ is given by $\mathcal{L}_{singleton}(\lambda, \theta; \mathcal{T}_t) + \sum_{j=1}^J \frac{1}{j!} \text{vec}[\partial^j L_t(\mathbf{c})]^\top \Delta^{\otimes j}$, where $\Delta = [\Delta_1^\top, \ldots, \Delta_K^\top]^\top$ and*

$$\Delta_k^\top = \mathbb{E}_{\substack{i \sim \mu_{t,k}, \\ j \sim \mu_{t', \sigma(k)}}} \left[ \alpha_{ij}^{(t,t')} \partial g \left( W \phi_{\theta_l}^l(\mathbf{x}_{t,i}^s) + b \right) W \left( \phi_{\theta_l}^l(\mathbf{x}_{t',j}^s) - \phi_{\theta_l}^l(\mathbf{x}_{t,i}^s) \right) \right].$$

To illustrate the effect of this data-dependent regularization, we now consider the following special case that is used by Yao et al. [50] for ProtoNet: $\mathcal{L}(\lambda, \theta; \mathcal{T}_t) = \frac{1}{n} \sum_{i=1}^n \mathcal{L}_i(\lambda, \theta; \mathcal{T}_t)$ where $\mathcal{L}_i(\lambda, \theta; \mathcal{T}_t) = \frac{1}{1 + \exp(\langle (\mathbf{x}_{t,i}^q - (\mathbf{c}_1' + \mathbf{c}_2')/2, \theta) \rangle)}$, $\mathbf{c}_k' := \frac{1}{N_{t,k}} \sum_{(\mathbf{x}_{t,i}^s, \mathbf{y}_{t,i}^s) \in \mathcal{D}_t^s} \mathbb{1}_{\{\mathbf{y}_{t,i} = k\}} \mathbf{x}_{t,i}^s$, and $\langle \cdot, \cdot \rangle$ denotes dot product. Define $c = \frac{1}{n} \sum_{i=1}^n \frac{1}{4} \frac{\psi(z_{t,i})(\psi(z_{t,i}) - 0.5)}{1 + \exp(z_{t,i})}$, where $\psi(z_{t,i}) = \frac{\exp(z_{t,i})}{1 + \exp(z_{t,i})}$ and $z_{t,i} = \langle \mathbf{x}_{t,i}^q - (\mathbf{c}_1' + \mathbf{c}_2')/2, \theta \rangle$. Note that $c > 0$ if $\theta$ is no worse than the random guess; e.g., $\mathcal{L}_i(\lambda, 0; \mathcal{T}_t) > \mathcal{L}_i(\lambda, \theta; \mathcal{T}_t)$ for all $i \in [n]$. We write $\|v\|_M^2 = v^\top M v$ for any positive semi-definite matrix $M$. In this special case, we consider that $\alpha$ is balanced: i.e., $\mathbb{E}_{t',\sigma}[\sum_{k=1}^2 \frac{1}{|I_{t',\sigma(k)}|} \sum_{j \in I_{t',\sigma(k)}} \frac{1}{|I_{t,k}|} \sum_{i \in I_{t,k}} \alpha_{ij}^{(t,t')}(\phi_{\theta_l}^l(\mathbf{x}_{t',j}^s) - \phi_{\theta_l}^l(\mathbf{x}_{t,i}^s))] = 0$ for all $t$. This is used to prevent the Set Transformer from over-fitting to the training sets; i.e., in such simple special cases, the Set Transformer without any restriction is too expressive relative to the rest of the model (and may memorize the training sets without using the rest of the model). The following proposition shows that the additional regularization term is simplified to the form of $c\|\theta\|_M^2$ in this special case:

**Proposition 1.** *In the special case explained above, the second approximation of $\mathbb{E}_{t',\sigma}[\mathcal{L}_{mix}(\lambda, \theta, \hat{\mathcal{T}}_{t,t'})]$ is given by $\mathcal{L}_{singleton}(\lambda, \theta; \mathcal{T}_t) + c\|\theta\|_{\mathbb{E}_{t',\sigma}[\delta_{t,t',\sigma} \delta_{t,t',\sigma}^\top]}^2$, where $\delta_{t,t',\sigma} = \mathbb{E}_{k \sim U[2]} \mathbb{E}_{i \sim \mu_{t,k}, j \sim \mu_{t',\sigma(k)}}[\alpha_{ij}^{(t,t')}(\mathbf{x}_{t',j}^s - \mathbf{x}_{t,i}^s)]$.*

In the above regularization form, we have an implicit regularization effect on $\|\theta\|_\Sigma^2$ where $\Sigma = \mathbb{E}_{\mathbf{x}, \mathbf{x}'}[(\mathbf{x} - \mathbf{x}')(\mathbf{x} - \mathbf{x}')^\top]$. The following theorem shows that this implicit regularization can reduce the Rademacher complexity for better generalization:

**Proposition 2.** *Let $\mathcal{F}_R = \{\mathbf{x} \mapsto \theta^\top \mathbf{x} : \|\theta\|_\Sigma^2 \leq R\}$ with $\mathbb{E}_{\mathbf{x}}[\mathbf{x}] = 0$. Then, $\mathcal{R}_n(\mathcal{F}_R) \leq \frac{\sqrt{R}\sqrt{\text{rank}(\Sigma)}}{\sqrt{n}}$.*

All the proofs are presented in Appendix A.

# 5 Experiments

We now demonstrate the efficacy of our set-based task augmentation method on multiple few-task benchmark datasets and compare against the relevant baselines.

**Datasets** We perform classification on eight datasets to validate our method. (1), (2), & (3) Metabolism [17], NCI [31] and Tox21 [18]: these are binary classification datasets for predicting the properties of chemical molecules. For Metabolism, we use three subdatasets for meta-training, meta-validation, and meta-testing, respectively. For NCI, we use four subdatasets for meta-training, two for meta-validation and the remaining three for meta-testing. For Tox21, we use six subdatasets

Table 1: Average accuracy of 5 runs and $\pm 95\%$ confidence interval for few shot classification on non-image domains – Tox21, NCI, GLUE-SciTail dataset, and ESC-50 datasets. ST stands for Set Transformer.

| | Chemical | | | Text | Speech |
| | Metabolism | Tox21 | NCI | GLUE-SciTail | ESC-50 |
| Method | 5-shot | 5-shot | 5-shot | 4-shot | 5-shot |
|---|---|---|---|---|---|
| ProtoNet | $63.62 \pm 0.56\%$ | $64.07 \pm 0.80\%$ | $80.45 \pm 0.48\%$ | $72.59 \pm 0.45\%$ | $69.05 \pm 1.48\%$ |
| MetaReg | $66.22 \pm 0.99\%$ | $64.40 \pm 0.65\%$ | $80.94 \pm 0.34\%$ | $72.08 \pm 1.33\%$ | $74.95 \pm 1.78\%$ |
| MetaMix | $68.02 \pm 1.57\%$ | $65.23 \pm 0.56\%$ | $79.46 \pm 0.38\%$ | $72.12 \pm 1.04\%$ | $71.99 \pm 1.41\%$ |
| MLTI | $65.44 \pm 1.14\%$ | $64.16 \pm 0.23\%$ | $81.12 \pm 0.70\%$ | $71.65 \pm 0.70\%$ | $70.62 \pm 1.96\%$ |
| ProtoNet+ST | $66.26 \pm 0.65\%$ | $64.98 \pm 1.25\%$ | $81.20 \pm 0.30\%$ | $72.37 \pm 0.56\%$ | $71.54 \pm 1.56\%$ |
| **Meta-Interpolation** | $\mathbf{72.92} \pm 1.89\%$ | $\mathbf{67.54} \pm 0.40\%$ | $\mathbf{82.86} \pm 0.26\%$ | $\mathbf{73.64} \pm 0.59\%$ | $\mathbf{79.22} \pm 0.84\%$ |

for meta-training, two for meta-validation, and four for meta-testing. (4) GLUE-SciTail [30]: it consists of four natural language inference datasets where we predict whether a hypothesis sentence contradicts a premise sentence. We use MNLI [47] and QNLI [46] for meta-training, SNLI [5] and RTE [46] for meta-validation, and SciTail [20] for meta-testing. (5) ESC-50 [34]: this is an environmental sound recognition dataset. We make a 20/15/15 split out of 50 base classes for meta-training/validation/testing and sample 5 classes from each spilt to construct a 5-way classification task. (6) Rainbow MNIST (RMNIST) [11]: this is a 10-way classification dataset. Following Yao et al. [50], we construct each task by applying compositions of image transformations to the images of the MNIST [9] dataset. (7) & (8) Mini-ImageNet-S [45] and CIFAR100-FS [22]: these are 5-way classification datasets where we choose 12/16/20 classes out of 100 base classes for meta-training/validation/testing, respectively and sample 5 classes from each split to construct a task.

Note that Metabolism, Tox21, NCI, GLUE-SciTail, and ESC-50 are real-world few-task meta-learning datasets with a very small number of tasks. For Mini-ImageNet-S and CIFAR100-FS, following Yao et al. [50], we artificially reduce the number of tasks from the original datasets for few-task meta-learning. RMNIST is synthetically generated by applying augmentations to MNIST.

**Implementation Details**  For RMNIST, Mini-ImageNet-S, and CIFAR100-FS, we use four convolutional blocks with each block consisting of a convolution, ReLU, batch normalization [19], and max pooling. For Metabolism, Tox21, and NCI, we convert the chemical molecules into SMILES format and extract a 1024 bit fingerprint feature using RDKit [15] where each bit captures a fragment of the molecule. We use two blocks of affine transformation, batch normalization, and Leaky ReLU, and affine transformation for the last layer. For GLUE-SciTail dataset, we stack 3 fully connected layers with ReLU on the pretrained language model ELECTRA [8]. For ESC-50 dataset, we pass raw audio signal to the pretrained VGGish [16] feature extractor to obtain an embedding vector. We use the feature vector as input to the classifier which is exactly the same as the one used for Metabolism, Tox21, and NCI. For our Meta-Interpolation, we use Set Transformer [23] for the set function $\varphi_\lambda$.

**Baselines**  We compare our method against following *domain-agnostic* baselines.

1. **ProtoNet** [40]: Vanilla ProtoNet trained on Eq. 2 by fixing $\varphi_\lambda$ to be the identity function.

2. **MetaReg** [2]: ProtoNet with $\ell_2$ regularization where element-wise coefficients are learned with bilevel optimization.

3. **MetaMix** [49]: ProtoNet trained with support sets and mixed query sets where we interpolate one instance from the support sets and the other from the original query sets with Manifold Mixup.

4. **MLTI** [50]: ProtoNet trained with Manifold Mixup based task augmentation. We sample two tasks and interpolate two query sets and support sets with Manifold Mixup, respectively.

5. **ProtoNet+ST** ProtoNet and Set Transformer ($\varphi_\lambda$) trained with bilevel optimization but without task augmentation for $\mathcal{L}_{\text{mix}}(\lambda, \theta, \hat{\mathcal{T}}_t)$ in Eq. 6.

6. **Meta-Interpolation** Our full model learning to interpolate support sets from two tasks using bilevel optimization and training the ProtoNet with both the original and interpolated tasks.

**Results**  As shown in Table 1, Meta-Interpolation consistently outperforms all the domain-agnostic task augmentation and regularization baselines on non-image domains. Notably, it significantly improves the performance on ESC-50, which is a challenging datatset that only contains 40 examples per class. In addition, Meta-Interpolation effectively tackles the Metabolism and GLUE-SciTail datasets which have an extremely small number of meta-training tasks: three and two meta-training tasks, respectively. Contrarily, the baselines do not achieve consistent improvements across all the

Table 2: Average accuracy of 5 runs and $\pm 95\%$ confidence interval for few shot classification on image domains — Rainbow MNIST, Mini-ImageNet, and CIFAR100. ST stands for Set Transformer.

| Method | RMNIST 1-shot | Mini-ImageNet-S 1-shot | 5-shot | CIFAR-100-FS 1-shot | 5-shot |
|---|---|---|---|---|---|
| ProtoNet | $75.35 \pm 1.43\%$ | $39.14 \pm 0.78\%$ | $51.17 \pm 0.57\%$ | $38.05 \pm 1.56\%$ | $52.63 \pm 0.74\%$ |
| MetaReg | $76.40 \pm 0.56\%$ | $39.36 \pm 0.45\%$ | $50.94 \pm 0.67\%$ | $37.74 \pm 0.70\%$ | $52.73 \pm 1.26\%$ |
| MetaMix | $76.54 \pm 0.72\%$ | $38.25 \pm 0.09\%$ | $52.38 \pm 0.52\%$ | $36.13 \pm 0.63\%$ | $52.52 \pm 0.89\%$ |
| MLTI | $79.40 \pm 0.75\%$ | $39.69 \pm 0.47\%$ | $52.73 \pm 0.51\%$ | $38.81 \pm 0.55\%$ | $53.41 \pm 0.83\%$ |
| ProtoNet+ST | $77.38 \pm 2.05\%$ | $38.93 \pm 1.03\%$ | $48.92 \pm 0.67\%$ | $38.03 \pm 0.85\%$ | $50.72 \pm 0.92\%$ |
| **Meta Interpolation** | $\mathbf{83.24} \pm 1.39\%$ | $\mathbf{40.28} \pm 0.48\%$ | $\mathbf{53.06} \pm 0.33\%$ | $\mathbf{41.48} \pm 0.45\%$ | $\mathbf{54.94} \pm 0.80\%$ |

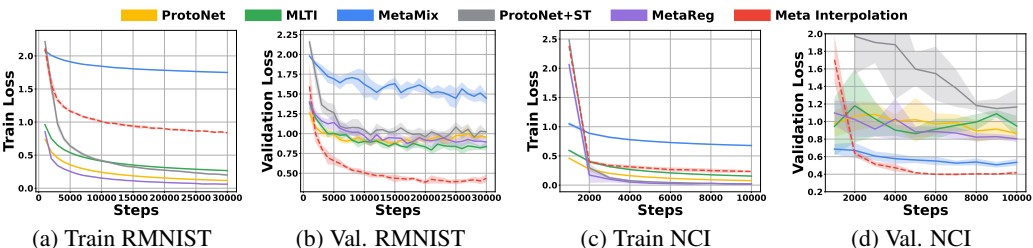

(a) Train RMNIST     (b) Val. RMNIST     (c) Train NCI     (d) Val. NCI

Figure 2: **(a)**∼**(d)** Meta-train and meta-validation loss on RMNIST and NCI for ProtoNet, MLTI, MetaMix, ProtoNet+ST, and Meta Interpolation.

domains and tasks considered. For example, MetaReg is effective on the sound domain (ESC-50) and Metabolism, but does not work on the chemical (Tox21 and NCI) and text (GLUE-SciTail) domains. Similarly, MetaMix and MLTI achieve performance improvements on some datasets but degrade the test accuracy on others. This empirical evidence supports the hypothesis that the optimal task augmentation strategy varies across domains and justifies the motivation for Meta-Interpolation which learns augmentation strategies tailored to each domain.

We provide additional experimental results on the image domain in Table 2. Again, Meta-Interpolation outperforms all the baselines. In contrast to the previous experiments, MetaReg hurts the generalization performance on all the image datasets except on RMNIST. Note that Manifold Mixup-based augmentation methods, MetaMix and MLTI, marginally improve the generalization performance for 1-shot classification on Mini-ImageNet-S and CIFAR-100-FS, although they boost the accuracy on 5-shot experiments. This suggests that different task augmentation strategies are required for 1-shot and 5-shot for the same dataset. Meta-Interpolation on the other hand learns task augmentation strategies tailored for each task and dataset and consistently improves the performance of the vanilla ProtoNet for all the experiments on the image datasets.

Moreover, we plot the meta-training and meta-validation loss on RMNIST and NCI dataset in Figure 2. Meta-Interpolation obtains higher training loss but much lower validation loss than the others on both datasets. This implies that interpolating only support sets constructs harder tasks that a meta-learner cannot memorize and regularizes the meta-learner for better generalization. ProtoNet overfits to the meta-training tasks on both datasets. MLTI mitigates the overfitting issue on RMNIST but is not effective on the NCI dataset where it shows high validation loss in Figure 2d. On the other hand, MetaMix, which constructs a new query set by interpolating a support and query set with Manifold Mixup, results in generating overly difficult tasks which causes underfitting on RMNIST where the training loss is not properly minimized in Figure 2a. However, this augmentation strategy is effective for tackling meta-overfitting on NCI where the validation loss is lower than ProtoNet. The loss curve of ProtoNet+ST supports the claim that increasing the model size and using bilevel optimization cannot handle the few-task meta-learning problem. It shows higher validation loss on both RMNIST and NCI as presented in Figure 2b and 2d. Similarly, MetaReg which learns coefficients for $\ell_2$ regularization fails to prevent meta-overfitting on both datsats.

Lastly, we empirically show that the performance gains mostly come from the task augmentation with Meta-Interpolation, rather than from bilevel optimization or the introduction of extra parameters with the set function. As shown in Table 1 and 2, ProtoNet+ST, which is Meta Interpolation but trained without any task augmentation, significantly degrades the performance of ProtoNet on Mini-ImageNet and CIFAR-100-FS. On the other datasets, the ProtoNet+ST obtains marginal improvement or largely

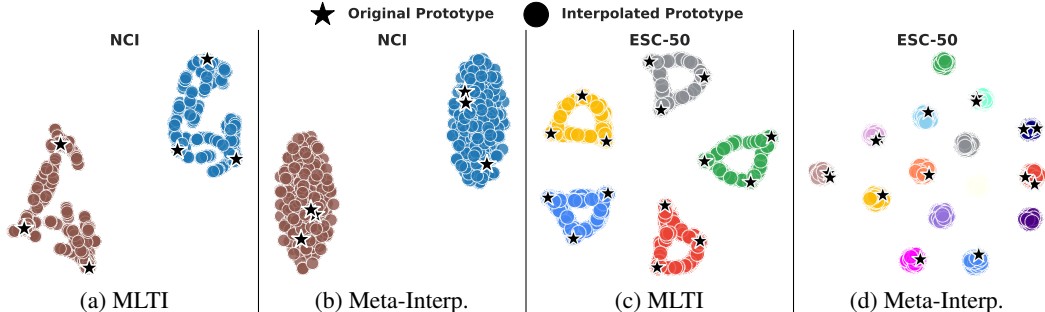

Figure 3: Visualization of original and interpolated tasks from NCI (**(a)** and **(b)**) and ESC-50 (**(c)** and **(d)**).

Table 3: Ablation study on ESC-50 dataset.

| Model | Accuracy |
|---|---|
| Meta-Interpolation | **79.22** $\pm$ 0.96 |
| w/o Interpolation | 71.54 $\pm$ 1.56 |
| w/o Bilevel | 63.01 $\pm$ 2.06 |
| w/o $\mathcal{L}_{\text{singleton}}(\lambda, \theta, \mathcal{T}_t^{\text{train}})$ | 78.01 $\pm$ 1.56 |

Table 4: Performance of different set functions on ESC-50 dataset.

| Set Function | Accuracy |
|---|---|
| ProtoNet | 69.05 $\pm$ 1.69 |
| DeepSets | 74.26 $\pm$ 1.77 |
| Set Transformer | **79.22** $\pm$ **0.96** |

Table 5: Performance of different interpolation on ESC-50 dataset.

| Interpolation Strategy | Accuracy |
|---|---|
| Query+Support | 76.87 $\pm$ 0.94 |
| Query | 78.19 $\pm$ 0.84 |
| Support+ Noise | 78.27 $\pm$ 1.24 |
| **Support** | **79.22** $\pm$ **0.96** |

underperforms the other baselines. Thus, the task augmentation strategy of interpolating two support sets with the set function $\varphi_\lambda$ is indeed crucial for tackling the few-task meta-learning problem.

**Ablation Study**  We further perform ablation studies to verify the effectiveness of each component of Meta-Interpolation. In Table 3, we show experimental results on the ESC-50 dataset by removing various components of our model. Firstly, we train our model without any task interpolation but keep the set function $\varphi_\lambda$, denoted as w/o Interpolation. The model without task interpolation significantly underperforms the full task-augmentation model, Meta-Interpolation, which shows that the improvements come from task interpolation rather than the extra parameters introduced by the set encoding layer. Moreover, bilevel optimization is shown to be effective for estimating $\lambda$, which are the parameters of the set function. Jointly training the ProtoNet and the set function without bilevel optimization, denoted as w/o Bilevel, largely degrades the test accuracy by $15\%$. Lastly, we remove the loss $\mathcal{L}_{\text{singleton}}(\lambda, \theta, \mathcal{T}_t^{\text{train}})$ for inner optimization in Eq. 6, denoted as w/o $\mathcal{L}_{\text{singleton}}(\lambda, \theta, \mathcal{T}_t^{\text{train}})$. This hurts the generalization performance since it decreases the diversity of tasks and causes inconsistency between meta-training and meta-testing, since we do not perform any interpolation for support sets at meta-test time.

We also explore an alternative set function such as DeepSets [52] using the ESC50 dataset to show the general effectiveness of our method regardless of the set encoding scheme. In Table 4, Meta-Interpolation with DeepSets improves the generalization performance of ProtoTypical Network and the model with Set Transformer further boost the performance as a consequence of higher-order and pairwise interactions among the set elements via the attention mechanism.

Lastly, we empirically validate our interpolation strategy that mixes only support sets. We compare our method to various interpolation strategies including one that mixes a support set with a zero mean and unit variance Gaussian noise. In Table 5, we empirically show that the interpolation strategy which mixes only support sets outperforms the other mixing strategies. Note that interpolating a support set with gaussian noise works well

Table 6: Comparison to interpolation with noise on ESC50.

| RMNIST | |
|---|---|
| **Interpolation Strategy** | **Accuracy** |
| Support+ Noise | 69.60 $\pm$ 1.60 |
| **Support** | **75.35** $\pm$ **1.63** |

on ESC50 dataset though we find that it significantly degrades the performance of ProtoNet on RMNIST, from $75.35 \pm 1.63$ to $69.60 \pm 1.60$ as shown in Table 6, which justifies our approach of mixing two support sets.

**Visualization**  In Figure 3, we present the t-SNE [43] visualizations of the original and interpolated tasks with MLTI and Meta-Interpolation, respectively. Following Yao et al. [50], we sample three original tasks from NCI and ESC-50 dataset, where each task is a two-way five-shot and five-way five-shot classification problem, respectively. The tasks are interpolated with MLTI or Meta-Interpolation to construct 300 additional tasks and represented as a set of all class prototypes. To visualize the

prototypes, we first perform Principal Component Analysis [13] (PCA) to reduce the dimension of each prototype. The first 50 principal components are then used to compute the t-SNE visualizations. As shown in Figure 3b and 3d, Meta-Interpolation successfully learns an expressive neural set function that densifies the task distribution. The task augmentations with MLTI, however, do not cover a wide embedding space as shown in Figure 3a and 3c as the mixup strategy allows to generate tasks only on the simplex defined by the given set of tasks.

## Limitation

Although we have shown promising results in various domains, our method requires extra computation for bilevel optimization to estimate $\lambda$, the parameters of the set function $\varphi_\lambda$, which makes it challenging to apply our method to gradient based meta-learning methods such as MAML. Moreover, our interpolation is limited to classification problem and it is not straightforward to apply it to regression tasks. Reducing the computational cost for bilevel optimization and extending our framework to regression will be important for future work.

## 6 Conclusion

We proposed a novel domain-agnostic task augmentation method, Meta Interpolation, to tackle the meta-overfitting problem in few-task meta-learning. Specifically, we leveraged expressive neural set functions to interpolate a given set of tasks and trained the interpolating function using bilevel optimization, so that the meta-learner trained with the augmented tasks generalizes to meta-validation tasks. Since the set function is optimized to minimize the loss on the validation tasks, it allows us to tailor the task augmentation strategy to each specific domain. We empirically validated the efficacy of our proposed method on various domains, including image classification, chemical property prediction, text and sound classification, showing that Meta-Interpolation achieves consistent improvements across all domains. This is in stark contrast to the baselines which improve generalization in certain domains but degenerate performance in others. Furthermore, our theoretical analysis shed light on how Meta-Interpolation regularizes the meta-learner and improves its generalization performance. Lastly, we discussed the limitation of our method.

## Acknowledgments

This work was supported by Institute of Information & communications Technology Planning & Evaluation (IITP) grant funded by the Korea government(MSIT) (No.2019-0-00075, Artificial Intelligence Graduate School Program(KAIST)), the Engineering Research Center Program through the National Research Foundation of Korea (NRF) funded by the Korean Government MSIT (NRF-2018R1A5A1059921), Institute of Information & communications Technology Planning & Evaluation (IITP) grant funded by the Korea government(MSIT) (No. 2021-0-02068, Artificial Intelligence Innovation Hub), the National Research Foundation of Korea (NRF) funded by the Ministry of Education (NRF-2021R1F1A1061655), Institute of Information & communications Technology Planning & Evaluation (IITP) grant funded by the Korea government(MSIT) (No.2022-0-00713), Samsung Electronics (IO201214-08145-01), and Google Research Grant. It was also results of a study on the "HPC Support" Project, supported by the 'Ministry of Science and ICT' and NIPA.

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
