# OpenReview forum: "Set-based Meta-Interpolation for  Few-Task Meta-Learning"
_NeurIPS.cc/2022/Conference — NeurIPS 2022 Accept_

### Official Review · Reviewer_w9NJ · 2022-07-09

**Rating:** 7
**Confidence:** 4
**Soundness:** 3 good
**Presentation:** 2 fair
**Contribution:** 3 good

**Summary:**

This paper presents a method for combining tasks in meta-learning when original base class domain data is scarce.  At meta-training time, a set transformer is used combine support set features from two random tasks, while comparing to queries from just the first.  The set transformer can therefore use the second random support set to apply random augmentations to the first, or not, as beneficial for the application.  Since the meta-training data is limited and easy to overfit, the random augmentations might never be used when trained jointly --- the transformer parameters are instead learned on a validation set with separate set of classes, using bilevel optimization with HyperGrad.  The method is evaluated on datasets in multiple domains, including chemical, text, speech and images, finding consistent improvements.


**Questions:**

Additional questions:

Since original base data is scarce to begin with, splitting out additional classes into a separate validation set to train the set transformer seems like a lot of data to take away from training the feature model parameters.  Is there a way to continue using these for both features and set transformer, maybe changing the held-out classes used in hypergrad periodically each outer loop?

How important is the second support set for use in the transformer?  What if random vectors are input to the set transformer instead of task 2 vectors?

Since there is just one set of set transformer params $\lambda$, does this limit the amount of augmentation or interpolation?  If the same data elements and permutations are used twice with same lambda params, the set transformer will output the same features both times, unless there is another source of randomness in the transformer that I missed.  Does this limit the approach at all?  Fig 3b shows a likely considerable amount of interpolation using several combinations of points, but could even more be generated from each pair using additional random sampling either of the transformer params or auxiliary input?


l.155 "interpolating either support or query sets achieves higher loss than interpolating both" --- reading this sentence alone (before sec 5), it's ambiguous if what is meant by "higher loss" is better or worse.  Reading this section, this question was very much on my mind, so seeing that it would be addressed by this sentence was great, but it wasn't clear which way it would be resolved.

alg.2:  some small typos and inconsistencies, lines 4,6 L_v suggests this is the loss but is already the grad, would be more consistent to accumulate a sum of losses in L_v and then set v_1 <- d/d\theta L_v.  also "single" is misspelled in the subscript



**Limitations:**

-

**Strengths And Weaknesses:**

This is an interesting approach that furthers work in task augmentation, extending beyond interpolation to learned mixing.  It's a little bit complex, using a set transformer, separate class split and bilevel optimization, and it's not entirely clear all components are necessary in their current form --- however, there are ablations showing each contributes, and particularly that a separate optimization is required.  I have a few questions below on how these might be able to be simplified, but also think the system is reasonable as-is to demonstrate the effectiveness of the approach.

There is also a theoretical section, but was very dense and a little outside my expertise; this claims to show a reduction in Rademacher complexity in simplified settings, but I didn't follow it in detail.  The end result of this section also seems to rely on reducing the upper bound by decreasing the rank of a covariance-like matrix of task vector differences, though no discussion on how or why its rank might be reduced.

Overall, the method is explained well and empirical results demonstrate its effectiveness, with particularly strong improvement in ESC-50.  The degree to which all components are needed in their current form, or whether they can be simplified, is unclear.  I would have have liked to see additional ablations on the form of the set transformer, random mixing vector selection, and validation set sizes.  But the paper does present ablations showing all components are indeed needed together at a high-level to justify the approach.

---

> ### Author Response · Authors · 2022-07-29
> **Response to Reviewer w9NJ**
>
> **[Q1]**. Theoretical section claims to show a reduction in Rademacher complexity in simplified settings. The end result of this section also seems to rely on reducing the upper bound by decreasing the rank of a covariance-like matrix of task vector differences, though no discussion on how or why its rank might be reduced.
>
> - Let $x=\phi(\bar x)$ where $\bar x \in \mathbb{R}^{\bar d}$ is the original data space and $x=\phi(\bar x) \in \mathbb{R}^{d}$ represents some features of $\bar x$. The full rank case corresponds to the situation where the data lives in the whole space of $\phi^{-1}(\mathbb{R}^d) \subseteq \mathbb{R}^{\bar d}$. In practice, we expect that the data lives in some (lower dimensional) manifold (instead of the whole space) where the rank is automatically reduced.
>
> ---
> **[Q2]**. Splitting out additional classes into a separate validation set to train the set transformer seems like a lot of data to take away from training the feature model parameters. Is there a way to continue using these for both features and set transformer, maybe changing the held-out classes used in hypergrad periodically each outer loop?
>
> - Thank you for your insightful suggestion. However, we need a separate meta-validation set for early stop as well as bilevel optimization. Moreover, we want to emphasize that the other baselines also require a separate validation set for early-stopping. Otherwise, they severely overfit to meta-training tasks since there are fewer tasks than conventional meta-learning scenario. Lastly, we use the same size of meta-validation set as all the baselines and our method does not require extra data.
> ---
> **[Q3]**. How important is the second support set for use in the transformer? What if random vectors are input to the set transformer instead of task 2 vectors?
>
> Thank you for insightful suggestion. As suggested, we interpolate support and gaussian random noise with zero mean and unit variance for task augmentation. Although mixing the support with the noise improves the performance of Prototypical Network on ESC50, it significantly degrades the test accuracy of Prototypical Network on RMNIST. In contrast, interpolating two support sets consistently boost the performance of the model, which shows the necessity of interpolating the two support sets.
>
>
> **[Ablation Study for Interpolation of two support sets ]**
>
> | Dataset | ProtoNet | Support1 + Noise | Support1 + Support |
> |----------------|----------------|:----------------:|:------------------:|
> | **RMNIST** | $75.35\pm1.63$ | $69.60\pm1.60$ | $\textbf{83.24}\pm\textbf{1.59}$ |
> | **ESC50** | $69.05\pm1.69$ | $78.27\pm1.24$ | $\textbf{79.22}\pm\textbf{0.96}$ |
> ---
> **[Q4]**. Since there is just one set of set transformer parameters $\lambda$, does this limit the amount of augmentation or interpolation? If the same data elements and permutations are used twice with same lambda parameters, the set transformer will output the same features both times, unless there is another source of randomness in the transformer that I missed. Does this limit the approach at all?
>
> - As we described the architecture of Set Transformer in Appendix D.3, **dropout** is the source of randomness to make output of Set Transformer stochastic. This is essential for generating diverse task augmentations, which explains how our Meta-Interpolation can generate a large number of diverse tasks given only three tasks, as shown in Figure 3b.
>
> ---
>
> **[Q5]**. "interpolating either support or query sets achieves higher loss than interpolating both" --- reading this sentence alone (before sec 5), it's ambiguous if what is meant by "higher loss" is better or worse.
>
>
> - By showing a higher **training loss**, we want to empirically support our intuition that interpolating only support sets makes the meta-training tasks harder so that meta-learner cannot easily memorize them.
>
> - It is hard to tell if a higher **training loss** is better since a meta-learner may underfit and show poor generalization. However, if the **training loss is higher** as a consequence of proper regularization, such as ours, the model achieves **lower validation loss** and better generalization performance as shown in Figure 2.
>
> ---
>
> **[Q6]**. Some small typos and inconsistencies, lines 4,6 $\mathcal{L}_V$ suggests this is the loss but is already the grad, would be more consistent to accumulate a sum of losses in $\mathcal{L}_V$ and then set $\mathbf{v}_1 \leftarrow \frac{\partial\mathcal{L}_V}{\partial \theta}$. also ``single" is misspelled in the subscript
>
> - Thanks for pointing out the typos and inconsistencies. For the inconsistency, we have corrected it as you suggested. For the typo, the notation $\mathcal{L}_\text{single}$ is inconsistent with $\mathcal{L}_\text{singleton}$ in Equations (2), so we have changed it to $\mathcal{L}_\text{singleton}$ in the revised version.

---

> > ### Comment · Reviewer_w9NJ · 2022-08-08
> > **responses**
> >
> > Thanks for the responses, they have mostly addressed my concerns, and I have raised my rating to 7.
> >
> > The random vectors experiment was also interesting, though I noticed the RMNIST result is only the in the text but not the table in the revision; I would recommend also adding it to the table in the final version, as it shows a nice contrast.

---

> > > ### Author Response · Authors · 2022-08-09
> > > **Thank you for your effort and time**
> > >
> > > Thank you for taking time and effort for carefully evaluating our responses. As suggested, we have included the RMNIST result in Table 16 in the revision. We thank you again for your helpful suggestions which significantly improved the quality of our paper.
> > >
> > > Thanks, Authors

---

### Official Review · Reviewer_Q5Gf · 2022-07-11

**Rating:** 6
**Confidence:** 3
**Soundness:** 3 good
**Presentation:** 3 good
**Contribution:** 4 excellent

**Summary:**

To tackle meta-overfitting in few-task meta-learning, the authors propose Meta Interpolation based on set transformer to generate domain-agnostic tasks. They assign a new class k by re-arrang K classes in two tasks, and the new class k is the k_{th} class in the two new permutations. Then, the corresponding support set is constructed with interpolating function using bilevel optimization. Since the interpolating function is optimized to minimize the loss on the validation tasks, the proposed method can achieve task augmentation for each specific domain. The effectiveness of Meta Interpolation is proved both experimentally and theoretically.

**Questions:**

1. Please analyse the principle of generating new classes by randomly aligning the category orders in two tasks. Since its prototype contains information of two classes, which may be quite different.
2. Why change the support set and keep the query set unchanged? If two new tasks contain the same query set but different support set, then, how to avoid the optimized support sets falling into the original one in task 1?
3.  Does this augmentation make task 1 and task 2 more similar?
4. More training details should be provided. For example, how many new tasks do you generate? Do the generated tasks make the comparison with other methods unfair (e.g., the model is updated more times during one training epoch, since more tasks are optimized in one epoch)? And which layer do you perform augmentation?
5. An ablation study about the number of training tasks and the performance of the proposed method, and the layer to implement augmentation may be better.

**Limitations:**

An ablation study about the number of training tasks and the performance of the proposed method may be better show the practical applicability.

**Strengths And Weaknesses:**

Pros:
1. The task of task generation is interesting.
2. The effectiveness of the proposed method is proved both experimentally and theoretically.

Cons:
1. The reasonability of using the he interpolating function to generate new support set, while keeping the query set unchanged is unclear.
2.  The way to generate new classes is somewhat hasty.

---

> ### Author Response · Authors · 2022-07-29
> **Response to Reviewer Q5Gf  (2/2)**
>
>
> **[Q2]**. Why are only the support sets interpolated but not the query sets?
>
>
> - As stated in line 152, the intuition behind interpolating support sets while keeping query sets intact is to make the task harder so that the meta-learner cannot easily memorize the tasks.
>
> - Our theoretical analysis support our intuition in that our interpolation strategy induces data-dependent regularization and improves the generalization bound of the meta-learner.
>
> - Lastly, in Table below, we empirically show that interpolation strategy which mixes only support sets outperforms the others.
>
>
>
>
> **[Accuracy for each interpolation strategy on ESC50]**
>
> | **Interpolation Strategy** | **Accuracy** |
> |--------------------------|:--------------------------------:|
> | Query | $78.19\pm0.84$ |
> | Query+Support | $76.87\pm0.94$ |
> | Support | $\textbf{79.22}\pm\textbf{0.96}$ |
>
>
>
> ---
>
>
> **[Q3]**. If two new tasks contain the same query set but different support set, then, how to avoid the optimized support sets falling into the original one in task 1?
>
> - The interpolated of the support sets cannot collapse to the original support set due to the following reasons. First, sampling exactly the same query sets is highly unlikely since we uniformly sample query sets from the full batch. Even if this happens, the hidden representations of two support sets are different due to the stochasticity introduced by the **dropout layer **  in the Set Transformer.
> ---
> **[Q4]**. Does this augmentation make task 1 and task 2 more similar?
>
> - This is a misunderstanding. We do not alter the original tasks with our proposed augmentation at all. Our method generates a new task given two tasks. Moreover, as shown in Figure 3, the prototypes of the different tasks are well separated, and does not show any sign of any prototypes being similar to each other.
>
> ---
> **[Q5]**. More training details should be provided. For example, how many new tasks do you generate? Do the generated tasks make the comparison with other methods unfair?
>
> - As shown in Algorithm 1, for each training step, we generate as many new tasks as batch size $B$, which is exactly the same batch size for all the other baselines.
>
> ---
> **[Q6]**. An ablation study about the number of training tasks and the performance of the proposed method, and the layer to implement augmentation may be better.
>
>
> - Thank you for the suggestion. First, we report how the test accuracy changes as we vary the number of meta-training tasks in the table below. The results show that our Meta-Interpolation consistently outperforms the baselines by large margins, regardless of the number of the tasks.
>
> **[Experiments on ESC50 dataset as varying the number of training tasks]**
>
> | **Model**              |            **5 Tasks**           |            **10 Tasks**           |            **15 Tasks**           |           **20 Tasks**           |
> |------------------------|:--------------------------------:|:---------------------------------:|:---------------------------------:|:--------------------------------:|
> | ProtoNet               |          $51.41\pm3.93$          |           $60.63\pm3.61$          |           $65.49\pm2.05$          |          $69.05\pm1.49$          |
> | MLTI                   |          $58.98\pm3.54$          |           $61.60\pm2.04$          |           $66.29\pm2.41$          |          $70.62\pm1.96$          |
> | **Meta-Interpolation** | $\textbf{72.74}\pm\textbf{0.84}$ | $\textbf{74.78}\pm\textbf{1.43}$  | $\textbf{77.47}\pm\textbf{1.33}$  | $\textbf{79.22}\pm\textbf{0.96}$ |
>
>
> -  We further report the test accuracies of our Meta Interpolation at different layers, in the table below.
>
> **[Accuracy  on ESC50 with different layers for interpolation]**
> | **Location of Interpolation** |           **Accuracy**           |
> |-----------------------------|:--------------------------------:|
> |          Input Layer          |          $66.83\pm1.31$          |
> |            Layer 1            |          $74.04\pm2.05$          |
> |            Layer 2            | $\textbf{79.22}\pm\textbf{0.96}$ |
> |            Layer 3            |          $77.62\pm1.46$          |
>
> ## Reference
> [1] Yao, Huaxiu, Linjun Zhang, and Chelsea Finn. "Meta-Learning with Fewer Tasks through Task Interpolation." International Conference on Learning Representations. 2021.

---

> ### Author Response · Authors · 2022-07-29
> **Response to Reviewer Q5Gf (1/2)**
>
> **[Q1]**. Please analyze the principle of generating new classes by randomly aligning the category orders in two tasks. Since its prototype contains information of two classes, which may be quite different.
>
>
>
> - Please recall that each task consists of its own classes for the K-way N-shot classification problem and our goal is to generate a new task by interpolating two classes.
>
>
>
> - If we have classes {$c_1, c_2, c_3$} and {$e_1, e_2, e_3$} for task1 and task2 respectively, we can construct a **new task** by assigning a new label for each combination {$c_1, e_2$} , {$c_2, e_3$}, and {$c_3, e_1$} and interpolate two instances from corresponding classes. For instance, we can mix an instance from the class $c_1$ and the one from the class $e_2$, and assign a new class {$c_1, e_2$} to the mixed example.
>
>
>
> - If we interpolate both support and queries with Manifold Mixup by following the above procedure, it is exactly how MLTI [1] creates new classes for task augmentation.
>
>
>
> - However, we observe that a meta-learner still overfit to the interpolated tasks as shown in Table below. Thus we propose to interpolate only support sets not query sets, which results in harder tasks that the meta-learner cannot easily memorize.
>
>
> - Since we do not interpolate query sets, there is no explicit prototype for the query set to be matched with. Instead, we match the query points from class $c_1$ with prototypes of {$c_1, e_2$} since the other prototypes do not contain any information of the class $c_1$.
>
>
> **[Accuracy for each interpolation strategy on ESC50]**
> | **Interpolation Strategy** | **Accuracy** |
> |--------------------------|:--------------------------------:|
> | Query | $78.19\pm0.84$ |
> | Query+Support | $76.87\pm0.94$ |
> | Support | $\textbf{79.22}\pm\textbf{0.96}$ |

---

> > ### Comment · Reviewer_Q5Gf · 2022-08-09
> > **Response**
> >
> > Thanks for the responses and clarification. Most of my concerns have been addressed, and I have raised my rating to 6.

---

### Official Review · Reviewer_7Fn3 · 2022-07-11

**Rating:** 6
**Confidence:** 4
**Soundness:** 3 good
**Presentation:** 3 good
**Contribution:** 3 good

**Summary:**

The paper introduces a new task augmentation method for meta-learning with few training tasks available. The proposed method employs a set-to-set function (instantiated as set transformer) that performs interpolation of features from the support set to create a new task. The parameters of a set transformer are learned via billevel optimization, using a meta-validataion dataset for tuning the parameters (considered as hyperparameters) to achieve the generalization. The paper performs experiments on the eight datasets to present the effectiveness of the proposed method, Meta-Interpolation.

**Questions:**

- What is the sensitivity of the proposed method to meta-validation dataset size, compared to other works, such as MLTI [48] that also performs feature interpolation?
- At which layer is the set transformer located? Is the location randomly sampled, similar to MLTI? If not, why so? Could you provide the ablation study on this?
- Why is the proposed method not applied to MAML? The similar work MLTI [48], which also performs feature interpolation, has shown the effectiveness on MAML as well. I believe providing experimental results on MAML will strengthen the paper.

**Limitations:**

While the paper provides limitations, I believe there are more limitations to discuss:
- the possible need for large meta-validation dataset (such high demand may break the assumption of few tasks).
- Validated only for prototypical networks

**Strengths And Weaknesses:**

### Strengths
- The paper presents extensive experimental results across diverse datasets.
- The idea of employing meta-learnable set-to-set function for feature interpolation is novel.

###  Weakness
- The number of parameters of the set transformer may be large enough that the proposed method may be sensitive to the size of meta-validation dataset. In fact, I noticed that the ratio of meta-training:meta-validation is quite high (e.g., 12:16 in the case of Mini-Imagenet-S).
- The proposed method is applied to only prototypical networks.

---

> ### Author Response · Authors · 2022-07-29
> **Response to Reviewer 7Fn3**
>
> **[Q1]**. How sensitive is the proposed method to meta-validation dataset size compared to other works?
>
> - Other than image datasets, as shown in Appendix D1, the size of the meta-validation set is extremely small (most of them consist of only two tasks). Thus our method is not sensitive to the size of the meta-validation set nor requires a large validation set, while being able to achieve significant performance improvements over the baselines.
>
> ---
> **[Q2]**. The proposed method is applied to only prototypical networks. Why is the proposed method not applied to MAML?
>
> - As we stated in the main paper, we focus solely on metric based meta-learning approaches (line 108-110) due to its efficiency and better empirical performance over the gradient based methods for the few-task meta-learning problem. The main challenge in applying our method to gradient-based meta-learning methods is that when our method is combined with bi-level gradient-based methods, it yields a tri-level optimization problem as it requires differentiating through second order derivative. The tri-lvel optimization is still known to be challenging problem [2, 3].
>
> -  However, it is possible to apply our method to first-order MAML [1] which approximate the Hessian with a zero matrix, and we provide additional experiments with this model on the ESC50 dataset. The experimental results show that Meta-Interpolation outperforms the relevant baselines, which shows the general applicability of our method. However, as mentioned before, the original Meta-Interpolation with metric-based meta-learning largely outperforms this model.
>
> **<Experiments on ESC50 dataset with first-order MAML>**
> |       **Method**       |           **Accuracy**           |
> |----------------------|:--------------------------------:|
> |   MAML w/ first-order  |          $72.14\pm 0.73$         |
> |   MAML w/ first-order  +     MLTI          |          $71.52\pm 0.55$         |
> | MAML w/ first-order + Meta-Interpolation| $76.68\pm1.02$ |
> | **ProtoNet + Meta-Interpolation** | $\textbf{79.22}\pm\textbf{0.96}$ |
>
>
> ---
> **[Q3]**. At which layer is the set transformer located? Is the location randomly sampled, similar to MLTI? If not, why so? Could you provide the ablation study on this?
>
> - As shown in Table 10 and 11 from Appendix D.4, we fix the location of interpolation. Otherwise we cannot use the same architecture of Set Transformer to interpolate output of different layers since hidden dimension of each layer is different. Moreover, we report the test accuracy for each layer to be interpolated. Interpolating hidden representation of support sets from layer 2, which is used in the main paper, achieves the best performance.
>
> **<Accuracy on ESC50 with different layers for interpolation>**
> | **Location of Interpolation** |           **Accuracy**           |
> |-----------------------------|:--------------------------------:|
> |          Input Layer          |          $66.83\pm1.31$          |
> |            Layer 1            |          $74.04\pm2.05$          |
> |            Layer 2            | $\textbf{79.22}\pm\textbf{0.96}$ |
> |            Layer 3            |          $77.62\pm1.46$          |
>
> ## References
> [1] Finn, Chelsea, Pieter Abbeel, and Sergey Levine. "Model-agnostic meta-learning for fast adaptation of deep networks." International conference on machine learning. PMLR, 2017.
>
> [2] Blair, Charles. "The computational complexity of multi-level linear programs." Annals of Operations Research 34 (1992).
>
> [3] Chen, Richard Li-Yang, Amy Cohn, and Ali Pinar. "An implicit optimization approach for survivable network design." 2011 IEEE network science workshop. IEEE, 2011.

---

> > ### Comment · Reviewer_7Fn3 · 2022-08-09
> > **Response to the authors**
> >
> > Thank you for the rebuttal, which addresses most of my concerns.
> > However, the rebuttal does not provide experiments on the sensitivity of the proposed method to the size of meta-validation.
> > Although the authors claim that the size of meta-validation is small for few datasets (NCI, Tox21, GLUE-SciTail), the ratio of meta-training:meta-validation is still high (e.g., 1:1 for GLUE-SciTail). And, other datasets (ESC-50, Mini-ImageNet, CIFAR-100-FS) still have a large number of meta-validation sets.
> > The fact that the proposed algorithm requires meta-validation set to tune hyperparameters, while the proposed method aims for meta-training with fewer number of tasks, still hints a gap that exists between the proposed method and the goal.
> > The experiments on how the size of meta-validation set affects the performance (esp. on datasets, such as Mini-ImageNet, that have a large number of meta-validation set) would be a nice touch that would lead to interesting discussions.
> > Regardless, the proposed method is novel and interesting. I have raised my rating to 6.

---

> > > ### Author Response · Authors · 2022-08-09
> > > **Reply to Reviewer 7Fn3**
> > >
> > > Thank you for taking the time and effort to review and re-evaluate our work. As suggested, we report how the test accuracy changes as we vary the number of **meta-validation tasks** in the table below and include it in the revision. Although the performance of Meta-Interpolation slightly decreases if we reduce the number of meta-validation tasks, it still outperforms the baselines by large margins, regardless of the number of meta-validation tasks.
> > >
> > >
> > > **[Test Accuracy on ESC50]**
> > > | Model              |              5 Tasks             |             10 Tasks             |    15 Tasks (full)    |
> > > |--------------------|:--------------------------------:|:--------------------------------:|:--------------:|
> > > | ProtoNet           |          $69.48\pm1.03$          |          $69.20\pm1.17$          | $69.05\pm1.49$ |
> > > | MLTI               |          $68.09\pm2.07$          |          $69.40\pm2.02$          | $70.62\pm1.96$ |
> > > | Meta-Interpolation | $\textbf{77.68}\pm\textbf{1.38}$ | $\textbf{77.13}\pm\textbf{1.23}$ | $\textbf{79.22}\pm\textbf{0.96}$ |

---

### Official Review · Reviewer_vsLv · 2022-07-23

**Rating:** 7
**Confidence:** 4
**Soundness:** 3 good
**Presentation:** 3 good
**Contribution:** 3 good

**Summary:**

The paper explores few-task meta-learning that attempts to ease the difficulty and cost of massive task construction. It challenges the existing few-task meta-learning approaches that heavily rely on the Manifold-Mixup-based task augmentation, which is hard to generalize to cases beyond computer vision. In this case, the author introduces a set-value neural net (Set transformer) as the tool to augment the tasks. The author provides a theorectical ground that Set transformer is able to control the Rademacher complexity for better generalization. 5 dataset and 3 image-based FS benchmarks have been used to evaluate the method.

**Questions:**

The paper may deserve a higher rating score if the questions below have been fully addressed.
1. Only Set-transformer has been considered as the implementation of meta-interpolation. Will other neural set functions be able to improve the few-task FS performance? It would be more convincing if the author can provide more empirical analysis on how to choose the set neural networks, e.g., change the architecture of the Set-transformer or use other types of set neural networks (for instance, Deep Set).
2. The method is claimed to be good at the field beyond vision. How is the performance in solving NLP low-resource problems?
3. Please refer to the first statement in limitation.

**Limitations:**

1. Distinct from the typical Mixup-based task augmentation baselines, the method is only implemented (only available?) in metric-based meta-learning, more precisely. ProtoNet. The author should provide more discussion of why meta-interpolation is limited in metric-based meta-learning, otherwise, provide the empirical performances of other meta-learning frameworks (e.g., MAML and more others) + meta-interpolation and further compare them with typical Mixup-based task augmentation baselines.
2. The method is a little bit incremental.

**Strengths And Weaknesses:**

1. The paper is well-organized and well-written.
2. The method seems simple but quite general and effective across many scenarios.
3. Theoretical results significantly motivate the method.
4. Massive empirical studies and the results are convincing.

---

> ### Author Response · Authors · 2022-07-29
> **Response to Reviewer vsLv (2/2)**
>
>
> **[Q3]**. The author should provide more discussion of why meta-interpolation is limited in metric-based meta-learning. Otherwise, provide experimental results for gradient based methods.
>
>
> - As stated in the main paper (line 108-110), we focus solely on metric based meta-learning  due to its efficiency and better empirical performance over the gradient based methods for the few-task meta-learning problem.
>
> - A crucial practical challenge in combining our method with the original MAML [1], is that it yields a tri-level optimization problem which requires differentiating through second order derivatives and the tri-level optimization still remains a **challenging** problem [2.3]. Thus, instead of using the original MAML, we provide additional experimental results on the ESC50 dataset using first-order MAML  which approximates the Hessian with a zero matrix. The experimental results show that Meta-Interpolation with first-order MAML outperforms MLTI, as shown below, which again confirms the general effectiveness of our set-based augmentation scheme. However, it largely underperforms our original Meta-Interpolation framework with metric-based meta-learning.
>
> **<Experiments on ESC50 dataset with first-order MAML>**
> |       **Method**       |           **Accuracy**           |
> |----------------------|:--------------------------------:|
> |   MAML w/ first-order  |          $72.14\pm 0.73$         |
> |   MAML w/ first-order  +     MLTI          |          $71.52\pm 0.55$         |
> | MAML w/ first-order + Meta-Interpolation| $76.68\pm1.02$ |
> | **ProtoNet + Meta-Interpolation** | $\textbf{79.22}\pm\textbf{0.96}$ |
>
>
> ---
> **[Q4]**. The method is a little bit incremental.
>
> - We do not believe that our method is incremental, since the use of an expressive set function allows to consider higher-order interactions among the set elements to create diverse augmentations that are off the simplex created by the highly **non-linear transformation** of the original samples, unlike Mixup-based strategies. Please See Figure 3 for the t-SNE visualization of the generated augmentations.  We believe that this scheme alone is highly novel and differs from any of the existing methods that simply interpolates between two instances.
>
> - Secondly, our interpolation strategy which generates augmentations only from the support sets, without using the query sets, is highly novel in the context of recent task augmentation strategies, as it makes tasks harder so that the meta-learner cannot easily memorize them.
>
> - Lastly, we provide non-trivial theoretical analysis showing that Meta-Interpolation induces data-dependent regularization and reduces the Rademacher complexity for better generalization.
>
> ## References
> [1] Finn, Chelsea, Pieter Abbeel, and Sergey Levine. "Model-agnostic meta-learning for fast adaptation of deep networks." International conference on machine learning. PMLR, 2017.
>
> [2] Blair, Charles. "The computational complexity of multi-level linear programs." Annals of Operations Research 34 (1992).
>
> [3] Chen, Richard Li-Yang, Amy Cohn, and Ali Pinar. "An implicit optimization approach for survivable network design." 2011 IEEE network science workshop. IEEE, 2011.

---

> > ### Comment · Reviewer_vsLv · 2022-08-08
> > **Response to Official Review of Paper3453 by Reviewer vsLv**
> >
> > The response has addressed my concerns. I would like to change the rating score to "7 Accept" of this work.

---

> > > ### Author Response · Authors · 2022-08-08
> > > **Thank you for your time and effort**
> > >
> > > We thank you for taking the time to carefully review our work and evaluate our responses. Could you reflect the updated score in the original review? We thank you again for your helpful suggestions which significantly improved the quality of our paper.
> > >
> > > Thanks, Authors

---

> > > > ### Comment · Reviewer_vsLv · 2022-08-09
> > > > **By the way, about the empirical studies for rebuttal**
> > > >
> > > > The reviewer noticed that the author did not include the MAML (implicit) in their paper, and the reviewer w9NJ found the same problem. It is probably because the author did not realize they had been authorized to revise their paper in the open review procedure. The author strongly encourages the authors to include a complete comparison with MAML (implicit) in table1,2,, since the extra empirical study is the main reason for the reviewer to raise this paper's rating.

---

> > > > > ### Author Response · Authors · 2022-08-09
> > > > > **Thank you for the suggestion**
> > > > >
> > > > > Thank you for your response and suggestion. We will include the experimental results in camera ready version since we have less than 12 hours for the interactive discussion. For the experiment on ESC dataset, we have included it on Appendix  F. We thank you again for your taking time and effort for the review.
> > > > >
> > > > > Thanks, Authors

---

> ### Author Response · Authors · 2022-07-31
> **Response to Reviewer vsLv (1/2)**
>
> **[Q1]**. Will neural set functions other than set transformer be able to improve the few-task few-shot performance? It would be more convincing if the author can provide more empirical analysis on how to choose the set neural networks.
>
> - We choose Set Transformer over DeepSets, since Set Transformer can model higher-order and pairwise interactions among the set elements via the attention mechanism while DeepSets uses a very simple aggregation and message-passing scheme, and thus obtains better empirical performance on various tasks. We report the accuracy of Meta-Interpolation with DeepSets in the table below to demonstrate this point.  DeepSets improves the performance of ProtoNet, which shows the general effectiveness of our method regardless of the set encoding scheme, but underperforms Set Transformer. We will include this new result in the revision to better justify the choice of the set encoder.
>
> **[Accuracy on ESC50 with different set functions]**
>
> | Set Function | Accuracy   |
> |----------------------|:--------------------------------:|
> |     ProtoNet     |  $69.05\pm 1.69$  |
> |     DeepSets     | $74.26\pm 1.77$ |
> |  SetTransformer  | $\textbf{79.22}\pm \textbf{0.96}$ |
>
>
> ---
> **[Q2]**. The method is claimed to be good at the field beyond vision. How is the performance in solving NLP low-resource problems?
>
> - Other than the vision and medical datasets, we also performed experiments for few-shot sentence classification using the GLUE-SciTail dataset. The results in Table 1 shows that our method outperforms all the Mixup-based augmentation methods, even on this NLP dataset.
>
>  **[Few-shot classification on GLUE-SciTail]**
> | Method                 |              Accuracy              |
> |------------------------|:----------------------------------:|
> | ProtoNet               |          $72.59 \pm 0.45$          |
> | MetaReg                |          $72.08 \pm 1.33$          |
> | MetaMix                |          $72.12 \pm 1.04$          |
> | MLTI                   |          $71.65 \pm 0.70$          |
> | ProtoNet + ST          |          $72.37 \pm 0.56$          |
> | **Meta-Interpolation** | $\textbf{73.64} \pm \textbf{0.59}$ |
> ---

---

### Author Response · Authors · 2022-07-31
**Summary of the Revision**

We really appreciate all the Reviewers for their constructive comments. We have responded to the individual comments from the Reviewers below, and believe that we have successfully responded to most of them. We have included the discussions and results of the suggested experiments in the revision. Here we briefly summarize the updates we have made to the revision:

- We have included additional experiments with first-order MAML on the ESC50 dataset  in Appendix F.1, as suggested by Reviewer vsLv and 7Fn3.

- We have included the experiments on the effects of the number of meta-training tasks in Appendix F.2, as suggested by Reviewer Q5Gf.

- We have performed ablation studies on the location of interpolation in Appendix F.3, as suggested by Reviewer 7Fn3 and Q5Gf.

- We have included the experiments with Deepsets in Appendix F.4, as suggested by Reviewer vsLv.

- We have performed experiments for different interpolation strategies including mixing  a support set and gaussian noise with zero mean and unit variance in Appendix F.5, as suggested by Reviewer Q5Gf  and w9NJ.

- We have corrected the typos in Algorithm1, as pointed out by Reviewer w9NJ.

- We have clarified the ambiguous sentence on line 156-157, as suggested by Reviewer w9NJ.

---

### Author Response · Authors · 2022-08-08
**A Gentle Reminder**

Dear Reviewers,

Could you please go over our responses and the revision since we can have interactions with you only by this Tuesday (9th 8:00 PM UTC)? We have responded to your comments and faithfully reflected them in the revision, and provided additional experimental results that you have requested. We sincerely thank you for your time and efforts in reviewing our paper, and your insightful and constructive comments.

Thanks, Authors

---

> ### Comment · Area_Chair_rHBH · 2022-08-09
> **I am asking the reviewers to engage with your responses**
>
> Dear authors,
>
> Thank you, I am asking the reviewers to engage with your responses before the deadline tomorrow. I sincerely hope that they will do so.
>
> Regards,
>
> AC

---

### Meta-Review · Area_Chair_rHBH · 2022-08-27

**Recommendation:** Accept
**Confidence:** Certain

**Metareview:**

This paper uses a set transformer to create new tasks at meta training time when the amount of data for meta training is scarce. This approach seems to be highly effective and will make a worthwhile contribution to the few-shot learning toolbox. Many of the reviewer concerns were addressed through additional ablations and experiments, e.g., using MAML instead of protonets. Please include these experiments in the final draft as they add quite a bit to the paper.

**Award:**

No

---

### Decision · Program_Chairs · 2022-09-14

Accept